# Generation and Characterization of Monoclonal Antibodies against Swine Acute Diarrhea Syndrome Coronavirus Spike Protein

**DOI:** 10.3390/ijms242317102

**Published:** 2023-12-04

**Authors:** Xinyue Zhou, Mengjia Zhang, Hanyu Zhang, Hailong Ma, Jiaru Zhou, Hua Cao, Guanghao Guo, Ningning Ma, Qigai He, Yongle Yang, Yifei Lang, Yaowei Huang, Wentao Li

**Affiliations:** 1National Key Laboratory of Agricultural Microbiology, College of Veterinary Medicine, Huazhong Agricultural University, Wuhan 430070, China; xinyuezhou0707@gmail.com (X.Z.); zhangmengjia0210@mail.hzau.edu.cn (M.Z.); pinkyy@webmail.hzau.edu.cn (H.Z.); littlelate@163.com (H.M.); jiaru@webmail.hzau.edu.cn (J.Z.); caohua@webmail.hzau.edu.cn (H.C.); guoguanghao@webmail.hzau.edu.cn (G.G.); mningning@webmail.hzau.edu.cn (N.M.); he628@mail.hzau.edu.cn (Q.H.); 2Key Laboratory of Prevention & Control for African Swine Fever and Other Major Pig Diseases, Key Laboratory of Development of Veterinary Diagnostic Products, Ministry of Agriculture and Rural Affairs, Wuhan 430070, China; 3Xianghu Laboratory, Hangzhou 311231, China; yang-yl@zju.edu.cn; 4College of Veterinary Medicine, Sichuan Agricultural University, Chengdu 611130, China; y_langviro@163.com; 5Guangdong Laboratory for Lingnan Modern Agriculture, College of Veterinary Medicine, South China Agricultural University, Guangzhou 510642, China; yhuang@zju.edu.cn; 6Hubei Hongshan Laboratory, Wuhan 430070, China

**Keywords:** monoclonal antibodies, neutralizing antibodies, swine acute diarrhea syndrome coronavirus, hemagglutination inhibition

## Abstract

Swine acute diarrhea syndrome coronavirus (SADS-CoV), a member of the family Coronaviridae and the genus Alphacoronavirus, primarily affects piglets under 7 days old, causing symptoms such as diarrhea, vomiting, and dehydration. It has the potential to infect human primary and passaged cells in vitro, indicating a potential risk of zoonotic transmission. In this study, we successfully generated and purified six monoclonal antibodies (mAbs) specifically targeting the spike protein of SADS-CoV, whose epitope were demonstrated specificity to the S1^A^ or S1^B^ region by immunofluorescence assay and enzyme-linked immunosorbent assay. Three of these mAbs were capable of neutralizing SADS-CoV infection on HeLa-R19 and A549. Furthermore, we observed that SADS-CoV induced the agglutination of erythrocytes from both humans and rats, and the hemagglutination inhibition capacity and antigen–antibody binding capacity of the antibodies were assessed. Our study reveals that mAbs specifically targeting the S1^A^ domain demonstrated notable efficacy in suppressing the hemagglutination phenomenon induced by SADS-CoV. This finding represents the first instance of narrowing down the protein region responsible for SADS-CoV-mediated hemagglutination to the S1^A^ domain, and reveals that the cell attachment domains S1^A^ and S1^B^ are the main targets of neutralizing antibodies.

## 1. Introduction

Swine acute diarrhea syndrome coronavirus (SADS-CoV), also known as porcine enteric alphacoronavirus (PEAV) [1] or swine enteric alphacoronavirus (SeACoV) [2], was the fourth porcine enteric coronavirus identified, following the documentation of porcine epidemic diarrhea coronavirus (PEDV), transmissible gastroenteritis coronavirus (TGEV), and porcine deltacoronavirus (PDCoV). SADS-CoV is a member of the genus Alphacoronavirus, family Coronaviridae, order Nidovirales [3,4].

Initially detected in 2016 in pig herds in Guangdong, China [1,2,5], SADS-CoV was believed to be a bat-origin coronavirus [5]. The symptoms of SADS-CoV-caused disease resemble those of other enteric coronaviruses, such as PDCoV and PEDV. Nevertheless, SADS-CoV presents a significant and undeniable threat to the swine breeding industry, as evidenced by the mortality of approximately 25,000 piglets during its initial outbreak in Qingyuan, Guangdong Province, China, spanning from 28 October 2016 to 2 May 2017, resulting in substantial economic losses for the industry [5]. In a retrospective survey conducted by Zhou et al., it was observed that the detection rate of SADS-CoV in 236 clinical samples of diarrhea, obtained from the Guangdong Province, China, between August 2016 and May 2017, amounted to 43.53%, indicating a noticeable trend of its prevalence in swine herds [6]. 

Coronaviruses are generally notorious for their rapid mutation and broad host tropism. Species tropism and susceptibility tests of SADS-CoV were conducted on 24 cell lines derived from different animals, including bats, mice, rats, guinea pigs, hamsters, pigs, chickens, and non-human primates. It was found that 21 of these cell lines demonstrated positive viral antigen expression and increased viral RNA titers after infection, indicating their sensitivity to SADS-CoV [7]. A similar investigation was conducted on another strain of SADS-CoV, and the selection of cell lines was supplemented with specific primary cell lines, the findings of which were consistent with the former report [8]. The susceptibility of rodent cell lines suggests that rodents are likely to be one of the natural intermediate hosts of SADS-CoV and may play an important role in the spread of the virus. Meanwhile, it has been discovered that SADS-CoV is also capable of infecting avian species [9]. A molecular analysis indicates that the SADS-CoV S protein may have originated from a hybrid infection involving both alpha- and beta-coronaviruses in bat reservoirs, which were subsequently transmitted to pigs [2]. The S protein of SADS-CoV exhibits high similarity to that of human coronavirus NL63 [10], while SADS-CoV could efficiently replicate in cells derived from various mammals, particularly exhibiting stable propagation in human cell lines [7,11]. This implies a high risk of cross-species transmission, especially to humans. 

Neutralizing antibodies against coronaviruses primarily target the spike (S) glycoprotein, which are present on the viral surface and play a crucial role in virus entry. The spike glycoprotein of coronaviruses mediates viral entry by binding to host cell receptors through the S1 subunit and facilitating fusion of the virus and cell membranes through the S2 subunit [12,13]. As shown in Figure 1, the S protein consists of two subunits responsible for distinct functions: the S1 subunit is composed of four core domains (S1^A^ through S1^D^) that facilitate cell attachment, while the S2 subunit mediates fusion between the viral and host cellular membranes [10]. Consequently, the S protein not only recognizes host receptors but also facilitates membrane fusion and internalization, making it a critical target for neutralizing antibody-mediated viral clearance [12,13]. Potent neutralizing antibodies typically bind to the receptor interaction site within the S1 subunit, inhibiting receptor binding [14]. Hence, usage of the S protein as a target for the development of coronaviral-neutralizing antibodies is ideal for maximum antiviral effect.

It was demonstrated that SADS-CoV does not rely on any known receptor of the other coronavirus upon cell entry [5], including human receptor angiotensin-converting enzyme 2 (ACE2) [15,16,17,18], dipeptidyl peptidase 4 (DPP4) [19], and aminopeptidase (APN) [17,20]. The virus’s ability to infect diverse cell types from different species [7,8] suggests the presence of a yet-undiscovered host receptor that is ubiquitous among mammals. Investigation into the interplay between SADS-CoV S and its host receptor shall offer valuable insights into the biology and pathogenesis of SADS-CoV and other coronaviruses.

The hemagglutination (HA) assay has extensive application in the detection and quantification of viruses that can agglutinate red blood cells. Certain members of the *Coronaviridae* family have demonstrated the ability to cause hemagglutination, including the other three known porcine intestinal coronaviruses [21,22,23,24,25]. The specific molecular mechanism underlying the hemagglutination activity of coronaviruses is believed to involve the interaction between the viral S protein and sialic acids (Sia). Additionally, investigations by Boschi et al. propose that the hemagglutination activity of SARS-CoV-2 arises from the binding between the N-terminal polysaccharide of the S protein and Sia glycoproteins abundantly present on the surface of human red blood cells [26]. This Sia -dependent adhesion may represent a significant molecular mechanism employed by animals for antiviral defense involving red blood cells and other blood cells [27,28]. Nonetheless, the formation of blood cell aggregates can lead to reduced oxygen-molecule binding capacity of red blood cells, endothelial damage, and hindered peripheral blood circulation, thereby exacerbating host tissue injury. Given that hemagglutination associated with coronaviruses may significantly impact viral infectivity and pathogenesis, evaluating the hemagglutination inhibition activity of antibodies becomes pivotal in assessing their potential as therapeutic agents. Currently, there is no evidence for the HA capability of SADS-CoV, highlighting the importance of addressing this issue.

Currently, there are no comprehensive studies on the S protein function, antibody neutralization activity, and hemagglutination inhibition activity of SADS-CoV. In order to address these gaps and provide further assistance to researchers in understanding SADS-CoV during their research and development of subunit vaccines and therapeutic drugs, this study prepared monoclonal antibodies specific to the S protein of SADS-CoV and investigated its characteristics.

## 2. Results

### 2.1. Generation and Characterization of SADS-CoV S-Specific mAbs

To identify the S-specific monoclonal antibodies (mAbs), a recombinant SADS-CoV S protein was expressed on the surface of HeLa-R19 cells (Figure 2a) and subjected to an immunofluorescence (IFA) assay using hybridoma cell culture supernatant. Six hybridoma cell lines (designated as mAbs 1#, 3#, 20#, 24#, 25#, and 32#) were shown to secrete antibodies against the SADS-CoV S protein. Furthermore, IFA was also performed on SADS-CoV-infected LLC-PK1 cells, and specific red fluorescence was observed in SADS-CoV-infected cells incubated with mAbs 1#, 3#, 20#, 24#, 25#, and 32# (Figure 2b). These results indicate that the six mAbs targeting the SADS-CoV S protein, namely 1#, 3#, 20#, 24#, 25#, and 32#, were successfully generated and capable of specific interaction with both recombinant and viral spike proteins.

The hybridoma cell lines secreting antibodies specifically targeting the SADS-CoV S protein then underwent three rounds of subcloning using limited dilution. Subsequently, the mAbs from ascites were purified with protein A beads. Further characterization using an isotyping kit revealed that all mAbs belonged to the IgG2/κ subclass.

### 2.2. Expression and Purification of Recombinant SADS-CoV S Protein

To further study the immunogenicity of the generated mAbs, recombinant proteins of the S1^A^ and S1^B^ domains were expressed using mammalian cell lines. SDS-PAGE and Western blotting (WB) suggested that the production of the recombinant proteins rS1^A^-hFc was approximately 60 kDa, and that of rS1^B^-hFc was approximately 50 kDa (Figure 3). Considering the protein glycosylation, the protein size was consistent with what was expected. These results indicate that the proteins rS1^A^-hFc and rS1^B^-hFc were successfully expressed and could be used for a subsequent enzyme-linked immunosorbent assay (ELISA).

### 2.3. Epitope Mapping of mAbs to the Different Domains of SADS-CoV Spike Protein

To identify the epitopes on different domains of the S protein of SADS-CoV, plasmids pCAGGS-SADS-S1-hFc, pCAGGS-SADS-S1^A^-hFc, pCAGGS-SADS-S1^B^-hFc, and pCAGGS-SADS-S2-hFc were transfected into HeLa-R19 cells, respectively (Figure 4a). IFA showed that mAbs 3#, 24#, and 25# exhibited specificity toward the S1^A^ domain, while mAbs 1#, 20#, and 32# were specific to the S1^B^ domain.

Additionally, rS1A-hFc and rS1^B^-hFc were coated and then linearized as rS1^A^-hFc-L and rS1^B^-hFc-L for ELISA, respectively. The ELISA results indicate that the epitopes recognized by mAb 24# were conformational, whereas the rest of them can target both the conformational and linear form of their target proteins (Figure 4b).

### 2.4. Binding of Purified Antibodies and SADS-CoV S Domains

By performing ELISA with rS1^A^-hFc and rS1^B^-hFc as the coating antigens, the EC_50_ values for S1^A^-specific mAbs 3#, 24#, and 25# were found to be 178.6, 12.56, and 118.10 ng/mL, respectively (Figure 5a). Similarly, the EC_50_ values for S1^B^-specific mAbs 1#, 20#, and 32# were determined to be 43.06, 18.41, and 21.67 ng/mL, respectively (Figure 5b and Table 1).

Biolayer interferometry (BLI) was employed to further examine the binding properties of the mAbs to SADS-CoV S domains. Initially, a preliminary experiment was conducted to determine the optimal concentration for loading the protein antigens rS1^A^-hFc and rS1^B^-hFc onto the protein A biosensor, which was found to be 5 µg/mL. Following a 2 min loading period, bovine serum was utilized for blocking. A subsequent washing step of 5 min was performed, followed by a 10 min association and a 4 min disassociation step. The binding kinetics of all antibodies to the SADS-CoV S1^A^ or S1^B^ proteins were evaluated. The results demonstrate that all antibodies exhibited high-affinity binding, as indicated by equilibrium dissociation constants (KDs) ranging from 2.84 × 10^−8^ to 1.67 × 10^−7^ (Figure 5c and Table 1).

These observations reaffirm the specific and stable binding capability of the mAbs prepared in this study to different SADS-CoV proteins. 

### 2.5. Neutralization Capacity of the mAbs In Vitro

To evaluate the neutralization capacity of the generated mAbs, neutralizing assays were performed on both the SADS-CoV-infected HeLa-R19 and A549 cell lines. mAbs 1#, 3#, 24#, and 25# exhibited potent inhibition of SADS-CoV infection in HeLa-R19 cells, with virus-neutralizing titers (VNT) of 62.5, 15.625, 62.5, and 15.625 µg/mL, respectively (Table 1). However, no neutralizing capability was observed for mAb 20# and 32#. The VNT values for antibodies 3#, 20#, 24#, and 25# in A549 cells were determined to be 31.25, 125, 15.625, and 125 µg/mL, respectively. Notably, the neutralizing capacity of mAbs 1# and 32# remained undetectable. Among the six tested monoclonal antibodies, mAb 24# demonstrated the most efficient neutralizing activity. The neutralizing capacity was also demonstrated on 3# as well as 20#. It seems that mAb 25# may hold a certain level of neutralizing efficacy, albeit relatively weaker in comparison.

### 2.6. Hemagglutination and Hemagglutination Inhibition Assay

To evaluate the potential hemagglutination inhibitory utility of the generated monoclonal antibodies (mAbs), hemagglutination assays with SADS-CoV were first conducted using human and rat erythrocytes, as shown in Figure 6. The results demonstrate that SADS-CoV exhibited the ability to induce the agglutination of both human and rat erythrocytes at 4 °C. The interaction between SADS-CoV and erythrocytes was found to be strictly dependent on Sia, as evidenced by the absence of hemagglutination when human and rat erythrocytes were pretreated with neuraminidase (NA) to remove Sia residues.

Subsequent analysis of hemagglutination inhibition showed that mAbs 3# and 25#, which specifically targeted S1^A^, effectively suppressed the hemagglutination of both erythrocytes (Figure 7). Conversely, mAbs targeting S1^B^ did not exhibit hemagglutination inhibitory activity. These findings suggest the presence of a crucial motif within S1^A^, which is responsible for the hemagglutinating activity of SADS-CoV. Thus, mAbs 3# and 25# appear as potential candidates for future investigations aiming at elucidation of the underlying mechanisms associated with this phenomenon.

## 3. Discussion

The generation of monoclonal antibodies is often hindered by the challenges posed by protein sequence variation and conformational changes resulting from engineering expression in eukaryotic or prokaryotic systems. The spike proteins of coronaviruses, with their intricate and crucial spatial structure, are highly vulnerable to hydrolysis during purification, further complicating the preservation of their native conformation when expressed in these systems. Consequently, the effective production of mAbs targeting the viral protein becomes arduous with such expression platforms.

To address these challenges, we adopted an alternative approach by utilizing SADS-CoV viral particles as immunogens to generate mAbs, circumventing the limitations associated with a eukaryotic or prokaryotic expression system. This approach offers several notable advantages, including reduced time and economic costs, enhanced immune response efficiency, mitigation of false-positive reactions arising from protein tagging or misfolding, and an improved yield of neutralizing antibodies. Employing this strategy, we successfully generated a panel of six monoclonal antibodies capable of effectively interacting with the SADS-CoV spike protein. Notably, to the best of our knowledge, this represents the first successful generation of monoclonal antibodies specifically targeting the SADS-CoV spike protein.

In the present study, we performed epitope mapping using recombinant proteins (rS1^A^-hFc, rS1^B^-hFc, rS1-hFc, and rS2-hFc) expressed in eukaryotic systems to delineate the specific domains targeted by the monoclonal antibodies. The interaction between the generated monoclonal antibodies and their corresponding recombinant proteins, rS1^A^-hFc and rS1^B^-hFc, was confirmed through ELISA and BLI assays, thus validating the epitope mapping results. However, determination of the precise epitopes on the SADS-CoV S protein that mediates the interaction of the mAbs is challenging due to the inherent susceptibility and intricate spatial structure of the spike protein, as previously discussed. Further elucidation of the epitopes may necessitate the application of techniques such as electron microscopy or hydrogen-deuterium exchange mass spectrometry (HDX-MS) to gain deeper insights.

Among the mAbs generated in this study, three of which exhibited prominent neutralizing activity against SADS-CoV. Notably, this is the first report of mAbs possessing neutralizing activity against SADS-CoV. Such activity was observed in mAbs targeting both the S1^A^ and S1^B^ domains, underscoring the pivotal role played by these domains in the infection process. The mouse-origin neutralizing antibodies generated through SADS-CoV viral particle immunization retained their neutralization efficacy when tested on human-derived cells. Given the ability of SADS-CoV to infect a diverse array of mammalian and avian cells in the absence of known coronavirus receptors, it is plausible that its receptor(s) are widely distributed across these animal species. The presence of neutralizing antibodies serves as a potent tool for elucidating the receptor(s) involved.

The Sia receptor plays a crucial role in the interaction between host cells and coronaviruses, even determining the tissue tropism and pathogenicity of the virus. However, no previous studies have demonstrated whether SADS-CoV interacts with Sia. The surface of red blood cells contains a substantial number of sialic acid receptors, and the HA test is frequently employed to verify the interaction between viruses and sialic acid receptors. In this study, for the first time, SADS-CoV was observed to interact with rat and human erythrocytes in the presence of Sia, leading to HA, which could be blocked by SADS-CoV S1^A^-specific mAbs. This finding supports the exiting conclusion that coronaviruses elicit hemagglutination through the S1 protein [26,29]. Our unpublished research revealed that SADS-CoV was unable to agglutinate chicken and pig erythrocytes and displayed a minimal degree of agglutination on mouse erythrocytes. Consequently, this study selected human erythrocytes and rat erythrocytes with higher hemagglutination values as test materials. It has been previously reported that PEDV also does not agglutinate chicken and pig erythrocytes, making this method suitable for differentiating PEDV from TGEV, which can agglutinate chicken erythrocytes [24] Hence, the difference between SADS-CoV and TGEV can also be observed by comparing the aggregation properties of red blood cells from different species. Reported studies suggest that both TGEV and PDCoV display Sia-dependent aggregation activity, which is similar to that of SADS-CoV [26,29]. In contrast, PEDV appears to be an exception [24]. The current research generally shows that the S protein, particularly the S1^A^ domain, interacts with Sia receptors, which is also consistent with the findings of this study. Therefore, we speculate that the mAbs 3# and 25# generated in this study might exert their hemagglutination-inhibiting effect by blocking the interaction between the S1^A^ domain of SADS-CoV and Sia. Given that hemagglutination by coronaviruses is likely a determining factor in the severity of clinical symptoms and tissue tropism, the mAbs 3# and 25# generated in this study have the potential to be developed as therapeutic agents in the future.

Numerous studies have reported the potential of SADS-CoV to exhibit a zoonotic tendency under in vitro cultivation conditions. Yang’s study demonstrated the ability of this virus to infect diverse cell types derived from pigs, monkeys, rats, mice, hamsters, chickens, bats, dogs, and humans [30]. Additionally, Edwards et al. provided further evidence illustrating the efficient replication of SADS-CoV in various human-derived cell lines originating from multiple organs in vitro [11]. Consistent with these investigations, our study verified the efficient infectivity of SADS-CoV in multiple animal cells, particularly those of human origin, with effective replication. The combination of the observed hemagglutination properties of SADS-CoV on human erythrocytes and the inhibitory effects of the generated mAbs on hemagglutination heightens suspicions regarding the zoonotic potential of SADS-CoV. However, there are currently no reported cases of human infection with SADS-CoV. Zhou et al. used their developed luciferase immunoprecipitation system to detect sera from 35 pig farmers who were in close contact with SADS-CoV-infected pigs, and no positive results were observed [5]. At present, it appears that the current strain of SADS-CoV poses a relatively low threat to human populations. Due to its potential to acquire higher infectivity and pathogenicity through mutation or recombination, the potential of SADS-CoV as a zoonotic pathogen still needs to be taken into account. Therefore, it is essential to strengthen the monitoring of SADS-CoV in both domestic and wild animal populations.

In conclusion, this study successfully generated a panel of six specific monoclonal antibodies (mAbs) targeting the S protein of SADS-CoV. Among them, mAbs 3#, 24#, and 25# were identified as binding specifically to the S1^A^ domain, while mAbs 1#, 20#, and 32# exhibited specific binding to the S1^B^ domain. Additionally, mAbs 3#, 20#, and 24# were confirmed to possess neutralizing capabilities against SADS-CoV. Moreover, this study demonstrated, for the first time, that SADS-CoV has the ability to agglutinate human and rat erythrocytes, and it confirms the hemagglutination inhibition activity of mAbs 3# and 25# against SADS-CoV. Notably, these hemagglutination inhibition activities were specific to the S1^A^ domain. These significant findings not only enhance our understanding of the antigenic functions of the S protein of SADS-CoV but also elucidate the specific interactions between SADS-CoV and host cells. Furthermore, these results provide valuable insights for the development of potential therapeutic approaches to combat SADS-CoV infection.

## 4. Materials and Methods

### 4.1. Cell Lines and Viruses

A mouse myeloma cell line (SP2/0, ATCC CRL-1581) was cultured in RPMI 1640 medium containing 20% fetal bovine serum (FBS) and 1% 100 μg/mL penicillin/streptomycin (Invitrogen). Vero (ATCC CRL-1587), LLC-PK1 (ATCC CL-101), HeLa-R19 (ATCC CRM-CCL-2) and A549 (ATCC CRM-CCL-185) were cultured in Dulbecco’s modified Eagle’s medium (DMEM, Invitrogen, Waltham, MA, USA) supplemented with 10% fetal bovine serum and 0.1% 100 μg/mL penicillin/streptomycin (Invitrogen). All cells were cultured at 37 °C in a 5% CO_2_ incubator. The SADS-CoV strain GDS04 (GenBank accession number: MF167434.1) used in this study was kindly provided by Professor Cao Yongchang from Sun Yat-sen University.

### 4.2. Plasmid Construction

Viral genomic RNA was isolated from the supernatant of SADS-CoV-infected Vero cells using TRIzol (Invitrogen). The extracted RNA was then used immediately for cDNA synthesis following the manufacturer’s instructions (Roche, Basel, Switzerland). Specifically, the full-length S1, S1^A^, S1^B^, and S2 domain sequences were amplified from the synthesized cDNA of SADS-CoV. These amplified sequences were subsequently cloned and inserted into pCAGGS-hFc vectors. The resulting recombinant plasmids, namely, pCAGGS-S1-hFc, pCAGGS-S1^A^-hFc, pCAGGS-S1^B^-hFc, and pCAGGS-S2-hFc, were confirmed through Sanger sequencing to ensure their accuracy and fidelity.

### 4.3. Eukaryotic Expression of S1^A^ and S1^B^ Domains

293T cells were suspended in culture medium at a density of 8 × 10^5^ cells/mL and subsequently transfected using PEI after 24 h of growth. The cells were transfected with plasmids pCAGGS-SADS-S1^A^-hFc and pCAGGS-SADS-S1^B^-hFc, encoding the recombinant proteins rS1^A^-hFc and rS1^B^-hFc for eukaryotic expression, and the cells were incubated at 37 °C for 6 days. Recombinant proteins were purified by protein A beads (GE Healthcare) according to the manufacturer’s instructions and verified by Western blot analysis.

### 4.4. Virus Amplification and Purification

SADS-CoV virus was propagated in Vero cells in the presence of trypsin. After achieving cytopathic effects (CPEs) of over 60%, the cell culture underwent three cycles of freeze—thaw at −80 °C. Subsequently, the post-freeze—thaw culture supernatant was subjected to centrifugation at 4 °C and 5000 rpm for 30 min, followed by filtration through a 0.45 μm filter. The filtered supernatant was then subjected to ultracentrifugation at 4 °C at 28,000 rpm. Following ultracentrifugation, the supernatant was discarded, and the pellet was resuspended in PBS. The purified virus was subsequently quantified for viral titers (TCID_50_) in Vero cells.

### 4.5. Immunization of Mice

Purified virus particles obtained through ultracentrifugation were utilized as antigens for immunizing mice in this experiment. Following each immunization, tail-tip blood samples were collected at two-week intervals, and antibody against SADS-CoV was subsequently valuated using an indirect ELISA. The mice with the highest antibody titers were selected for spleen cell harvesting and cell fusion.

Female BALB/c mice (6 weeks of age) were obtained from the Experimental Animal Institution of Huazhong Agricultural University and housed in specific pathogen-free (SPF) isolated cages under negative-pressure ventilation. For the immunization, each 6-week-old BALB/c mouse was immunized with 10^8^ TCID_50_ SADS-CoV emulsified in complete Freund’s adjuvant (Sigma, St. Louis, MO, USA). Subsequently, three rounds of booster immunizations were performed at 3-week intervals, except that the virus was emulsified in incomplete Freund’s adjuvant. The mice with higher serum antibody titers were intraperitoneally injected with 2.5 × 10^7^ TCID_50_ of SADS-CoV without adjuvant three days prior to cell fusion.

Animal care and all experimental procedures were conducted following ethical guidelines and approved protocols. The animal experiments were approved by the Animal Ethics Committee of Huazhong Agriculture University, with the approval number HZAUMO-2023-0209.

### 4.6. Enzyme-Linked Immunosorbent Assay

NUNC Maxisorp plates were coated with rS1^A^-hFc and rS1^B^-hFc at a concentration of 50 ng/well and incubated overnight at 4 °C. The plates were then washed three times with phosphate-buffered saline (PBS) containing 0.05% Tween-20 and blocked with 5% skim milk in PBS containing 0.1% Tween-20 at room temperature for 2 h. Tenfold serial dilutions of mAbs, starting at a concentration of 1 µg/µL, were added to the plates and incubated at room temperature for 1 h. After three washes, the plates were incubated with HRP-conjugated goat anti-mouse secondary antibody (ABclonal, Woburn, MA, USA), diluted 1:10,000, for one hour at room temperature. The absorbance values were measured at OD_630nm_ using tetramethylbenzidine (TMB) substrate and an ELISA plate reader.

### 4.7. Immunofluorescence Assay

IFA were conducted to detect monoclonal antibodies specific to different domains of the S protein. Briefly, LLC-PK1 cells infected with SADS-CoV or plasmid-transfected HeLa-R19 cells were washed with phosphate-buffered saline (PBS) and fixed with 4% paraformaldehyde. Subsequently, the cells were permeabilized using 0.1% Triton X-100, followed by incubation with appropriate hybridoma supernatants and secondary antibodies and staining with 4’,6-diamidino-2-phenylindole (DAPI). Fluorescence images were acquired using a fluorescence microscope (SOPTOP ICX41, Suzhou, China).

### 4.8. Virus Neutralization Assay

In brief, monoclonal antibodies were serially diluted twofold in culture medium, starting at a concentration of 250 µg/mL, while 50 µL of each dilution was mixed with 50 µL of SADS-CoV (100 TCID_50_) for 1 h at room temperature. The mixture was then added to HeLa-R19 and A549 cells and incubated for 2 h at 37 °C. Following this, the cells were washed and further incubated in medium for 8 h at 37 °C. Subsequently, the cells were fixed and stained using a mouse anti-SADS-CoV-N monoclonal antibody (generated by our laboratory) followed by secondary antibody Alexa Fluor 594 donkey anti-mouse IgG. The results were verified using a fluorescence microscope (SOPTOP ICX41).

### 4.9. Binding Kinetics and Affinity Measurements

The binding kinetics and affinity of monoclonal antibodies (mAbs) to the SADS-CoV-S domains were assessed using biolayer interferometry (BLI). The optimal loading concentration of recombinant SADS-CoV-S domains onto anti-human Fc biosensors (Pall ForteBio, Fremont, CA, USA) was determined beforehand to ensure non-saturation of the sensor. The kinetic binding assay involved loading the recombinant SADS-CoV proteins onto the biosensor at the optimal concentration (5 µg/mL) for 2 min. Following this, the sensor was incubated with a range of mAb concentrations (50–0.78 µg/mL, twofold dilution) for 200 s to allow for antigen association. A dissociation step in PBS for 4 min was then executed after the association step. The kinetic constants were calculated using a 1:1 Langmuir binding model with ForteBio Data Analysis software (version 11.1.0.4).

### 4.10. Hemagglutination Assay

Human and rat erythrocytes (Huizhi Heyuan Biotechnology Co., Ltd., Beijing, China) were washed three times with PBS. In some experiments, these erythrocytes were pretreated for 3 h at 37 °C with NA. Serial 2-fold dilutions of SADS-CoV or influenza A virus (IAV, strain PR8, accession number: LC120389.1) were incubated with erythrocytes in V-bottom, 96-well plates for 30 min at 4 °C. Subsequently, the hemagglutination titer was determined. For the hemagglutination inhibition assay, SADS-CoV (8 hemagglutinating unit, HAU) was pretreated or mock-pretreated with serial 2-fold dilutions of monoclonal antibodies (starting at 1 µg/µL) at 4 °C for 1 h, followed by the addition of human or rat erythrocytes.

## Figures and Tables

**Figure 1 ijms-24-17102-f001:**
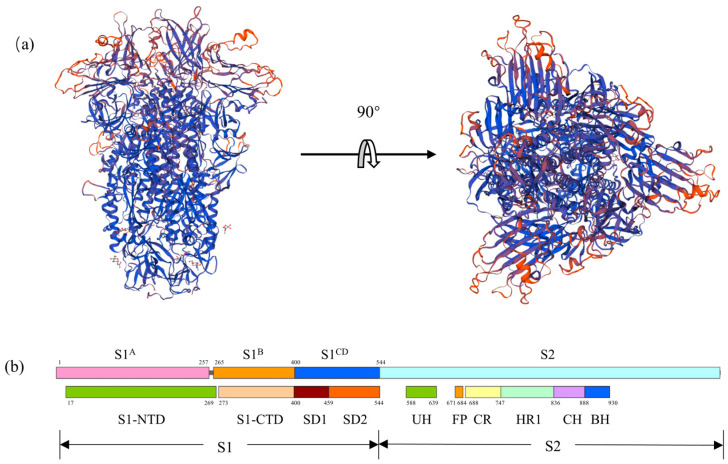
Three-dimensional modeling (**a**) and domain arrangement (**b**) of the S protein of SADS-CoV.

**Figure 2 ijms-24-17102-f002:**
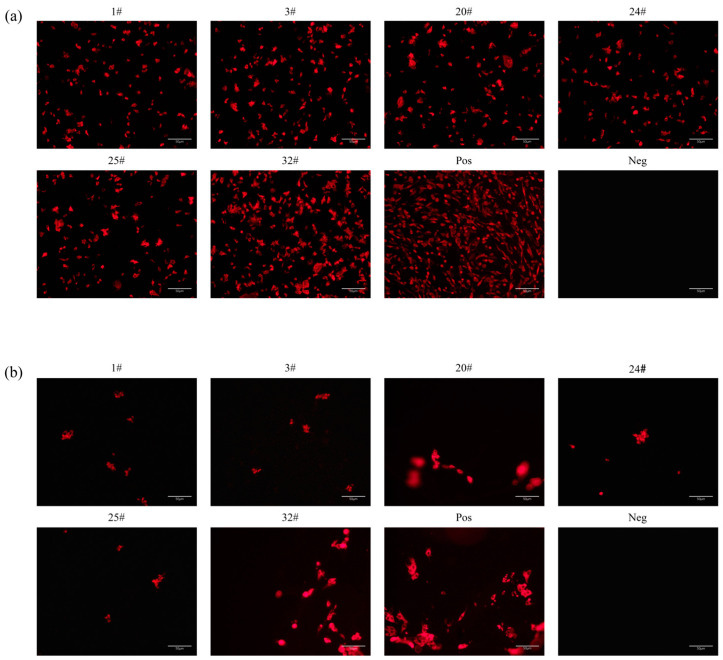
Screening of SADS-CoV spike-specific monoclonal antibodies. (**a**) The pCAGGS-SADS-S-flag plasmid was transfected into HeLa cells, and the resultant culture supernatant was utilized as a source of antibodies for the immunofluorescence assay. Immunized mouse serum was used as a positive control, and negative mouse serum was used as a negative control. Scale bar in the pictures were 50 μm. (**b**) LLC-PK1 cells were infected with SADS-CoV for 24 h. Hybridoma supernatant was used as the primary antibody in the immunofluorescence assay. Immunized mouse serum was used as a positive control, and negative mouse serum was used as a negative control. Scale bar in the pictures were 50 μm.

**Figure 3 ijms-24-17102-f003:**
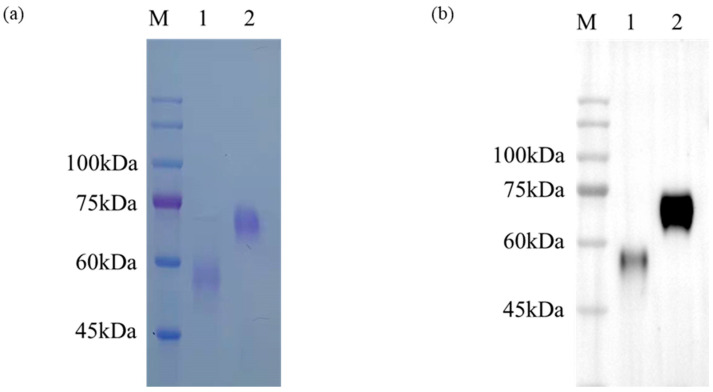
Eukaryotic expression of rS1^A^-hFc and rS1^B^-hFc verified by SDS-PAGE (**a**) and Western blot (**b**) with HRP goat anti-human IgG. 1. Purified rS1^B^-hFc protein, whose expected size is 45.4 kDa; 2. Purified rS1^A^-hFc protein, whose expected size is 56.0 kDa; M, represents protein marker.

**Figure 4 ijms-24-17102-f004:**
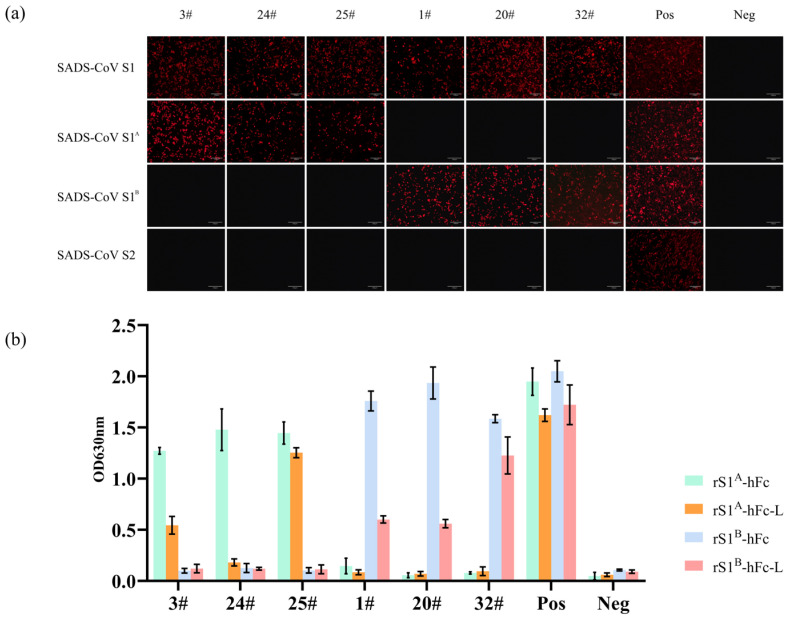
Characterization of SADS-CoV S reactive antibodies. (**a**) Epitopes of purified mAbs were mapped. rS1-hFc, rS1^A^-hFc, rS1^B^-hFc, and rS2-hFc were expressed in HeLa cells and analyzed with purified antibodies. Immunized mouse serum was used as a positive control, and negative mouse serum was used as a negative control. Scale bar in the pictures were 100 μm. (**b**) ELISA was performed to distinguish conformational or linear epitopes. rS1^A^-hFc and rS1^B^-hFc were coated on the plates, and half of them were pretreated with sodium lauroyl sarcosinate (SKL) to linearize the proteins, while the others were mock-pretreated. Immunized mouse serum was used as a positive control, and negative mouse serum was used as a negative control.

**Figure 5 ijms-24-17102-f005:**
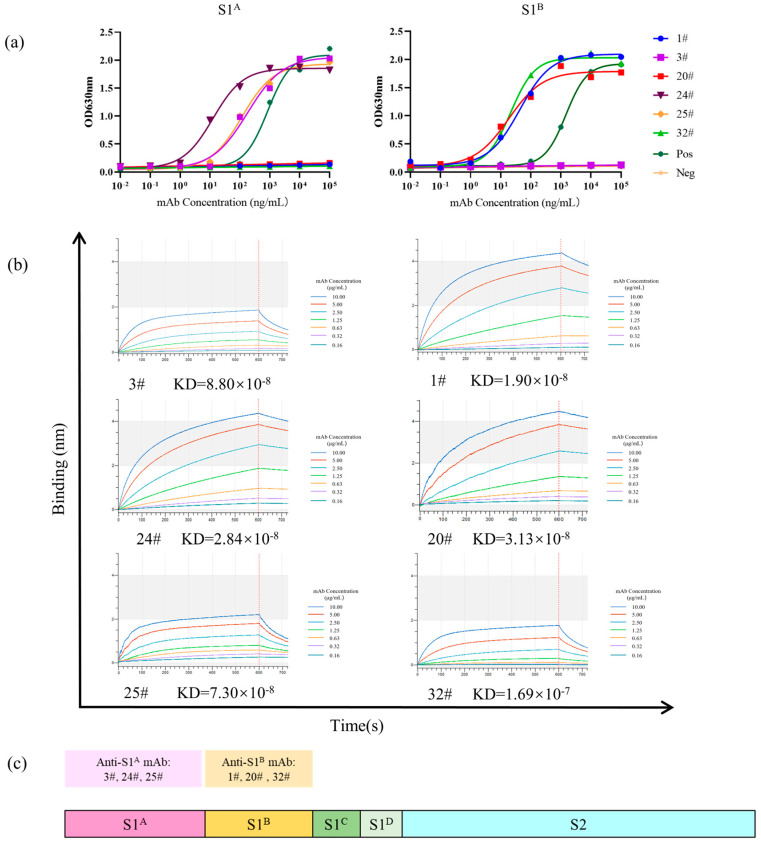
Binding of purified antibodies and SADS-CoV S domains. (**a**) The EC_50_ values were determined by ELISA reactivity of purified mAbs to rS1^A^-hFc and rS1^B^-hFc. (**b**) The binding affinity between purified monoclonal antibodies and target proteins was determined using a biolayer interferometry (BLI) assay, and the KD values were quantitatively calculated. (**c**) Schematic distribution of epitope groups of anti-SADS-S mAbs over the different SADS-S domains.

**Figure 6 ijms-24-17102-f006:**
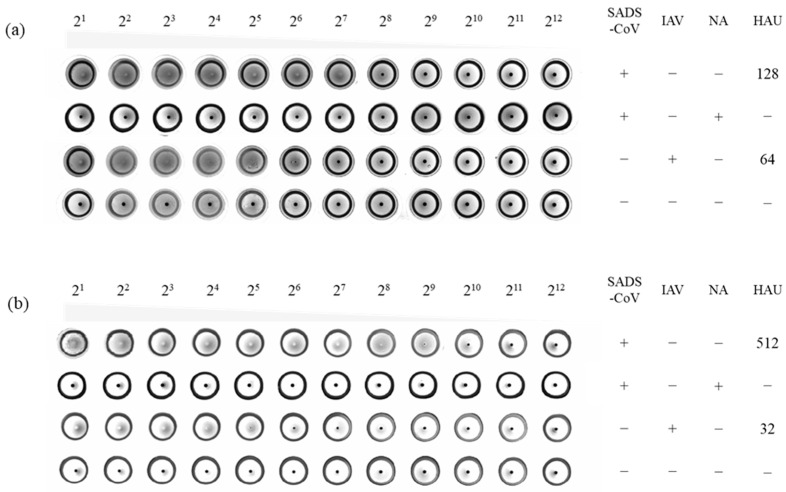
SADS-CoV exhibited hemagglutination capacity on rat (**a**) and human (**b**) erythrocytes relying on Sia-containing receptors on erythrocyte surface. Human erythrocytes were mock-treated (PBS) or NA-treated, and incubated with a two-fold serial dilution of SADS-CoV. Influenza A virus was used as a positive control and PBS was used as a negative control.

**Figure 7 ijms-24-17102-f007:**
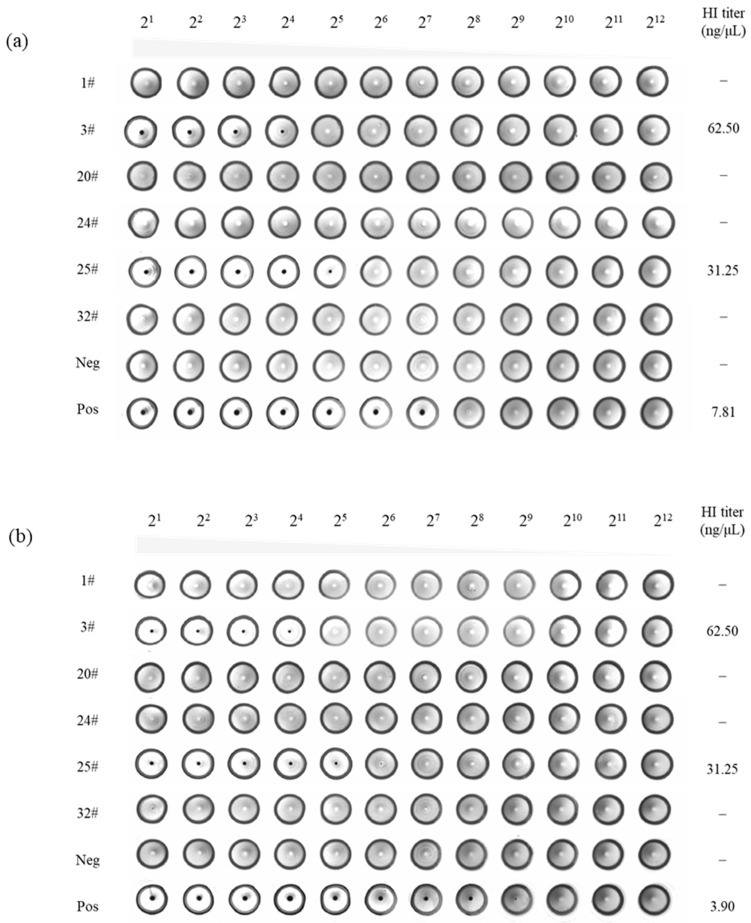
Hemagglutination inhibition assay on rat (**a**) and human (**b**) erythrocytes. After incubation at 4 °C for 30 min, purified monoclonal antibodies (mAbs) and 8 HAU of SADS-CoV were separately reacted with human and rat erythrocytes. Immunized mouse serum was used as a positive control and negative mouse serum was used as a negative control.

**Table 1 ijms-24-17102-t001:** Characterization of the generated mAbs against the SADS-CoV S protein.

mAb	Isotype	Binding Region						EC50 (ng/mL)	KD (M)	Neutralizing Titer (μg/mL)		Hemagglutination Inhibition Titer (μg/mL)	
		SADS-CoV	S	S1	S1A	S1B	S2			HeLa-R19	A549	Rat	Human
1#	IgG2a/κ	+	+	+	−	+	−	43.06	1.90 × 10^−8^	62.50	NA	NA	NA
3#	IgG2b/κ	+	+	+	+	−	−	178.6	8.80 × 10^−8^	15.63	31.25	62.50	62.50
20#	IgG2a/κ	+	+	+	−	+	−	18.41	3.13 × 10^−8^	62.50	125.00	NA	NA
24#	IgG2a/κ	+	+	+	+	−	−	12.56	2.84 × 10^−8^	15.63	15.63	NA	NA
25#	IgG2a/κ	+	+	+	+	−	−	118.1	7.30 × 10^−8^	125.00	125.00	31.25	31.25
32#	IgG2b/κ	+	+	+	−	+	−	21.67	1.69 × 10^−7^	NA	NA	NA	NA

## Data Availability

All data generated and analyzed during this study are included in this published manuscript.

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
