# Peer review of "Generation and Characterization of Monoclonal Antibodies against Swine Acute Diarrhea Syndrome Coronavirus Spike Protein"

_ijms, 2023, doi:10.3390/ijms242317102_

Round 1
Reviewer 1 Report
Comments and Suggestions for Authors
Zhou et al generated monoclonal antibodies against SADS-CoV and successfully utilized ELISA and immunofluorescence assay to map the epitope to the S1 region of the spike protein. Further characterization of the antibodies revealed that 3 of the mAbs were neutralizing while 3 were non-neutralizing mAbs. This is an interesting study. The manuscript is properly organized and easy to understand.
However, I am of the opinion that generation and characterization of mAbs alone do not constitute studies that meet any of the criteria included in the scope of IJMS as stated below:
- Fundamental theoretical problems of broad interest in biology, chemistry and medicine;
- Breakthrough experimental technical progress of broad interest in biology, chemistry and medicine;
- Application of the theories and novel technologies to specific experimental studies and calculations.
Author Response
Thank you for your valuable suggestions. We would like to highlight that this manuscript has been submitted to the IMJS Special Issue "Molecular Research on Coronavirus: Pathogenic Mechanism, Antiviral Drugs, and New Vaccines," which aligns perfectly with the focus of this issue.
Furthermore, we have come across a related article titled "Identification of a Monoclonal Antibody against Porcine Deltacoronavirus Membrane Protein," authored by Huiguang Wu et al and published in IMJS earlier this year. This article shares similarities with our work, indicating that there is considerable interest in this field among the editor and readers. We aim to contribute our research to this interested audience.
In our study, we have successfully confirmed the Sia-dependent hemagglutination capacity of SADS-CoV. Moreover, our results suggested that the S1A domain might play a crucial role in this process. Additionally, our study represents the first identification of neutralizing monoclonal antibodies against SADS-CoV. These findings offer valuable insights for future investigations into the structure and function of SADS-CoV S protein.
Reviewer 2 Report
Comments and Suggestions for Authors
I recommend making the following changes to the manuscript.
1. Line 57: I think I would change sensitivity to susceptibility.
Lines 165-170: I think this is an oversimplification of what's actually happening with 25 and 32. Some antibodies can truly only target linear epitopes, the conformational form of the protein actually has no reactivity. In this case, 25 & 32 can target both a conformational and a linear. Therefore, I don't think it's fair to call them "linear" Abs, instead they can target both.
Lines 208-219: Ab 25 was one of your most potent inhibitors in HeLa cells, and yet a poor neutralizer in A549 cells. Similarly 1 neutralized in HeLa cells, but not in A549s. I would like some additional comment/clarification on this. Does this suggest different parts of the spike protein are critical for different cell lines? Different receptors are being used? Please expand on this finding.
Line 226: Please add rat as well. Both human and rat erythrocytes are dependent on sialic acid
Comments on the Quality of English Language
While I appreciate that English is not the authors first language, a reasonable amount of english changes are needed throughout the document. I have highlighted several here.
Line 39: Change is to was (past tense).
Line 46: Change resemble to resembling
Line 47: Change an unneglectable to "its own serious"
Line 59: please add the so it reads "consistent with the former report"
Line 71: please change plays to play.
Line 89: please change use to uses
Line 119: please delete "shall to"
Line 135: please replace "were" with "then"
Author Response
- Line 57: I think I would change sensitivity to susceptibility.
The word has been replaced as suggested.
- Lines 165-170: I think this is an oversimplification of what's actually happening with 25 and 32. Some antibodies can truly only target linear epitopes, the conformational form of the protein actually has no reactivity. In this case, 25 & 32 can target both a conformational and a linear. Therefore, I don't think it's fair to call them "linear" Abs, instead they can target both.
We greatly value the significance of this suggestion and sincerely appreciate your advice. Consequently, we have taken this suggestion into serious consideration and have made revisions to the mentioned section in our manuscript. However, we would like to note that mAb 24# exhibited limited interaction with rS1A-hFc-L, leading us to believe that the epitope for this particular monoclonal antibody remains linear.
- Lines 208-219: Ab 25 was one of your most potent inhibitors in HeLa cells, and yet a poor neutralizer in A549 cells. Similarly 1 neutralized in HeLa cells, but not in A549s. I would like some additional comment/clarification on this. Does this suggest different parts of the spike protein are critical for different cell lines? Different receptors are being used? Please expand on this finding.
We appreciate your observation and believe that the variation in mAb neutralizing activity between different cell lines can be attributed to the varying abundance of receptors rather than SADS-CoV binds to different cell types. This phenomenon is similar to what has been observed in COVID-19 cases, where variations in clinical symptoms between men and women have been linked to differences in the content of ACE2 receptors.
Previous reports have demonstrated that SADS-CoV can infect various animal lineages and primary cells, including bats, rodents, non-human primates, birds, and humans. This suggests that the receptors utilized by SADS CoV are likely to be widely distributed among these species, rather than relying on multiple different receptors for invasion. It is important to note that the utilization of different receptors would require infection and adaptive mutations in different animals, for which there is currently no reported clinical evidence.
Furthermore, SADS-CoV has shown high sensitivity to various respiratory cells, such as A549, indicating the presence of unknown receptor proteins in respiratory cells that can be recognized by the virus and aid in its infection. As a cervical cancer cell line, HeLa cells exhibit significant functional differences from respiratory cell lines, which could explain the differential receptor abundance observed.
When comparing the neutralizing activity of various antibodies, we have observed differences in neutralization potency between different cell lines. However, the overall trend remains consisten. There is no evidence to suggest that antibody A has a stronger neutralizing ability than antibody B on HeLa cells, while antibody B exhibits stronger neutralizing ability than antibody A on A549 cells. This consistency in trend leads us to believe that SADS CoV employs the same receptor(s) when invading these two cell types, rather than utilizing different invasion mechanisms.
- Line 226: Please add rat as well. Both human and rat erythrocytes are dependent on sialic acid
The word has been added as suggested.
Comments on the Quality of English Language
All the words mentioned have been changed as recommended. However, some words have been deleted or replaced due to modifications to the article content.
Reviewer 3 Report
Comments and Suggestions for Authors
The manuscript “Generation and Characterization of Monoclonal Antibodies against Swine Acute Diarrhea Syndrome Coronavirus Spike Protein” was carefully analyzed. The authors investigated different monoclonal antibodies to neutralize SADS-CoV. The preliminary results seem to be interesting, but some sections should be reviewed in this work.
Introduction:
Lines 46-48: More information regarding the economic impact of this virus should be indicated (it is25,000 piglets... where? For how long it was recorded?, etc)
Lines 49-51: the 2nd highest infection rate compared to?
Lines 85-113: I suggest summarizing these lines/paragraphs, because different human and animal virus are mentioned and this information is not related with the main objective of this paper.
Lines 115 to 121: The objective is not well defined. This paragraph could fit more in conclusion section than introduction.
Material and Methods/Results:
Why do you use hemagglutination if RBC are not the cell target for the virus? Why do you perform this analysis in human and rat and no in pigs? Is this virus considered as zoonotic agent? In this case, such point should be noted in the introduction because its control is even higher.
Discussion/Conclusion:
Preliminary results obtained in this study are poorly discussed, these are only described (and some parts/messages of the introduction are repeated). I kindly invite to the authors to compare their results with other swine coronaviruses in order to potentiate this interesting and novel work.
Author Response
- Lines 46-48:More information regarding the economic impact of this virus should be indicated (it is25,000 piglets... where? For how long it was recorded?, etc)
Thank you for your inquiry. We have added further details to lines 46-50 of the manuscript as suggested.
- Lines 49-51: the 2nd highest infection rate compared to?
In this retrospective study, conducted between August 2016 and May 2017, it was observed that among 236 diarrheal samples, SADS-CoV had the second highest positivity rate, only surpassed by PEDV. These details have been added to the manuscript on lines 50-53.
- Lines 85-113: I suggest summarizing these lines/paragraphs, because different human and animal virus are mentioned and this information is not related with the main objective of this paper.
We greatly appreciate these suggestions, and we have made the necessary changes on lines 88-113.
- Lines 115 to 121: The objective is not well defined. This paragraph could fit more in conclusion section than introduction.
In response to this valuable suggestion, we have taken the initiative to replace the paragraph with a new one in an effort to better convey the meaning of our research. Details can be seen on lines 115-120.
- Why do you use hemagglutination if RBC are not the cell target for the virus? Why do you perform this analysis in human and rat and no in pigs? Is this virus considered as zoonotic agent? In this case, such point should be noted in the introduction because its control is even higher.
Erythrocyte surface contains a large number of sialic acid receptors. In the study conducted by Krempl et al titled "Characterization of the sialic acid binding activity of transmissible gastroenteritis coronavirus by analysis of haemagglutination-deficient mutants", a haemagglutination assay was used to investigate the interaction between TGEV and sialic acid receptors. Hence, in order to determine whether SADS-CoV also interacts with sialic acid receptors, it is imperative to perform a haemagglutination test. In the preliminary test, apart from human and rat red blood cells, our study also attempted to utilize red blood cells from mice, chickens, and pigs for haemagglutination tests. However, the latter did not demonstrate any haemagglutination. The inability to agglutinate pig red blood cells is also observed in cases of PEDV and PDCoV.
In their study titled "Fatal swine acute diarrhoea syndrome caused by an HKU2-related coronavirus of bat origin," Zhou et al. examined SADS-CoV antigen in blood samples taken from pig farm workers, but no presence of the virus was detected. Currently, there have been no reports of SADS-CoV infection in humans, and most scholars consider it as a potential risk for future zoonotic transmission, although it has not been confirmed as a zoonotic disease.
As an RNA virus, coronaviruses frequently undergo recombination or mutation. If the broad host tropism of SADS-CoV were passed to a new zoonotic coronavirus through recombination or mutation, it could potentially pose a significant threat to the livestock industry and human health. Therefore, the potential risk of SADS-CoV to humans should not be overlooked.
- Preliminary results obtained in this study are poorly discussed, these are only described (and some parts/messages of the introduction are repeated). I kindly invite to the authors to compare their results with other swine coronaviruses in order to potentiate this interesting and novel work.
The constructive suggestion has been taken into account. In lines 280-301, we have further discussed and compared the phenomenon of HA in our study, while also addressing and supplementing the content of question 5.
Round 2
Reviewer 1 Report
Comments and Suggestions for Authors
The authors have adequately addressed my concern.
Author Response
Thanks for your positive reponse.